# *Semliki Forest* Virus (SFV) Self-Amplifying RNA Delivered to J774A.1 Macrophage Lineage by Its Association with a Purified Recombinant SFV Capsid Protein

**DOI:** 10.3390/ijms25147859

**Published:** 2024-07-18

**Authors:** Roselane P. Gomes, Flavia F. Barbosa, Marcelo A. S. Toledo, Soraia A. C. Jorge, Renato M. Astray

**Affiliations:** 1Viral Biotechnology Laboratory, Butantan Institute, Av. Vital Brasil 1500, São Paulo 05503-900, Brazil; roselane.gomes@fundacaobutantan.org.br (R.P.G.); soraia.jorge@butantan.gov.br (S.A.C.J.); 2Programa Interunidades em Biotecnologia, Universidade de São Paulo, São Paulo 05508-060, Brazil; flavia.barbosa@fundacaobutantan.org.br; 3Multipurpose Laboratory, Butantan Institute, Av. Vital Brasil 1500, São Paulo 05503-900, Brazil; 4Department of Hematology, Oncology, Hemostaseology and Stem Cell Transplantation, RWTH Aachen University Medical School, 52074 Aachen, Germany; halbmund@gmail.com; 5Center for Integrated Oncology Aachen Bonn Cologne Düsseldorf (CIO ABCD), 52074 Aachen, Germany

**Keywords:** alphavirus replicon, capsid protein, Semliki Forest virus

## Abstract

The Semliki Forest virus capsid protein (C) is an RNA binding protein which exhibits both specific and unspecific affinities to single-strand nucleic acids. The putative use of the self-amplifying RNAs (saRNAs) of alphaviruses for biotechnological purpose is one of the main studied strategies concerning RNA-based therapies or immunization. In this work, a recombinant C protein from SFV was expressed and purified from bacteria and used to associate in vitro with a saRNA derived from SFV. Results showed that the purified form of C protein can associate with the saRNA even after high temperature treatment. The C protein was associated with a modified saRNA coding for the green fluorescent protein (GFP) and delivered to murine macrophage cells which expressed the GFP, showing that the saRNA was functional after being associated with the recombinant purified C protein.

## 1. Introduction

Genetic vaccines are the third generation on the vaccine technology development field [1]. Besides their capacity of generating balanced humoral and cellular immune responses, genetic vaccines generally present several advantages over protein-based vaccines (subunit or whole organism), such as low production costs, rapid development, standardized production process and high stability [2,3]. Recently, mRNA vaccines have proven their value as one of the most important instruments for the control of the COVID-19 pandemic [4]. Aiming to improve the immunogenicity and lowering nucleic acid doses, many studies have been conducted for the development of biolistic particle delivery system, clinical electroporation devices, liposome or nanoparticle formulations and viral capsid-based delivery [2]. Another strategy for the use of decreased amounts of nucleic acid material is by delivering replicon-based vaccines [5], with some recent candidates reaching clinical trials as COVID-19 vaccine candidates [6,7]. After being introduced in a cell, the replicon or self-amplifying RNA (saRNA) can generate thousands of copies of an mRNA coding for a specific antigen, and consequently, a high number of antigen molecules is produced [8,9]. *Alphavirus* saRNAs have long been studied for vaccination [10]; recently, bi-cistronic saRNAs for bivalent vaccines [11] and transreplicons where RNA polymerase and the RNAs of interest are delivered separately, requiring only 50 ng for mouse immunization [12], have shown potential improvements in the field. A possible strategy to deliver saRNAs involves the use of an *Alphavirus* replication-defective vector. However, due to several concerns related to the safety of this system, the possibility of existent immunity against the vector as well as the costs and difficulties in its production impaired the development of an *Alphavirus* replicon-based vaccine candidate. Good alternatives have been proposed for saRNA delivery [5,13], including lipid nanoparticles (LNPs), liposomes and polymeric nanoparticles made from biocompatible polymers such as poly(lactic-co-glycolic acid), as well as viral vectors and lipid–polymer hybrid complexes. Among these, LNPs are particularly notable for their efficiency in encapsulating and protecting mRNA and can be engineered to enhance stability and facilitate uptake by cells. The choice of delivery system depends on factors like the specific mRNA vaccine, target cell types stability requirements and desired immune response [14]. The capsid protein of *Semliki Forest* virus (C), an old-world *Alphavirus*, is a 33 kDa protein containing 267 amino acids which associates with the positive strand self-amplifying RNA (180:1 copies) in viral nucleocapsids [15]. This property of associating to RNA molecules is both unspecific and specific [16]. The first region of the protein (amino acids 1 to 80), which presents a concentration of basic aminoacids, especially lysine, provides the unspecific association between the C protein and RNA molecules or even single-strand DNA molecules [17]. The specific association is based on the packing signal present in the non-structural RNA segment, which was particularly well described for the activity of the *Sinbdis* virus C protein [18]. More recently, it was shown that the packing of virus genomic RNA depends on the binding of the capsid protein to multiple high affinity sites, favoring specific recognition of the viral RNA and its packaging into new viral particles [19]. In this work, we produced the capsid protein of *Semliki Forest* virus in bacteria and used it to build a capsid-like structure containing an *Alphavirus* saRNA expressing the green fluorescent protein (GFP), and used it to deliver the RNA package to cells in culture. The recombinant capsid protein showed preserved property of associating to the saRNA-GFP even after high temperature incubation prior to the association.

## 2. Results

The plasmid containing the C protein gene under the control of T7 promotor was inserted into *E. coli* BL21 (DE3), and expression was induced with IPTG at 37 °C, 30 °C or 28 °C. Cell pellets of different clones were collected and lysed by sonication for protein analysis (Figure 1). Results of immunoblotting analysis showed the successful expression of a 35 kDa histidine-tagged C protein and revealed that the expression at 28 °C was the best choice regarding protein amount. In all cases, the presence of a lower molecular weight histidine-tagged product around 20 kDa was noticed. This truncated form of the protein was present several times after induction (0 h to 6 h) and at different IPTG concentrations (0.01 mM or 0.5 mM) or even using other *E. coli* strains (Appendix A).

The clone 1 was selected for further experiments of purification and RNA association because of the improved production at 28 °C and lower production of the 20 kDa truncated protein respective to clone 3. The clone was inoculated into 250 mL of culture medium, and expression was induced for 6 or 3 h for protein production at 28 °C or 37 °C, respectively, to check the quality of the protein produced in terms of the property of association with the saRNA-GFP. For both cultures, the cell pellet was lysed by sonication and clarified, and the cell extract was submitted to IMAC for C protein purification. Results of the purification step for the protein produced at 28 °C are shown (Figure 2A).

After the first purification step, the major part of the protein was obtained in the fraction eluted with 800 mM imidazole. In this fraction, the greatest part of the smaller product was also eluted. In the step using 1200 mM imidazole, a relatively pure C protein of the expected size was found. The fraction containing the C protein eluted at 1200 mM imidazole was then submitted to a desalting step for buffer change and additional removal of contaminants (Figure 2B).

When the fraction of C protein was used for RNA–protein association and analyzed by gel shift assay, it was noticed that the C protein was likely associated with trace amounts of contaminating nucleic acids, as the lane loaded with C protein alone stained with nucleic acid GelRed^TM^ dye, especially in the case of the protein produced at 28 °C (Figure 3, lane 1). When the protein was produced at 37 °C, the amount of contaminating nucleic acids was shown to be lower (Figure 3, lane 6). The C protein also showed the property of associating with the MAYV_E2 RNA. Although this RNA codes for an *Alphavirus* protein, it does not have a packing signal identified in its sequence, confirming the property of C protein to associate with other RNA sequences than the replicon.

Considering that the same bacterial clone was used for the production of the C protein in the two temperatures and the protein produced at 37 °C was less contaminated with nucleic acids, the incubation of C protein at high temperatures was tested as a step in the protein preparation before protein and saRNA-GFP association; we hypothesized that the high temperature would break the unspecific association (Figure 4). In this experiment, although the C protein produced was relatively pure, as no nucleic acid labeling was identified in the purified material, the contamination with nucleic acid cannot be excluded considering the sensitivity of the assay (>50 ng for DNA fragments depending on the length). The purified C protein was then treated at 70 °C or 100 °C for 10 min to prevent any remaining nucleic acids and protein associations and subsequently mixed with saRNA-GFP at 30 °C. The association of saRNA-GFP and C protein after high temperature incubation was confirmed in the mobility shift assay by blocking saRNA-GFP migration through the agarose gel (Figure 4).

The association of saRNA-GFP and C protein was then analyzed by transmission electron microscopy (TEM). Both C protein incubated at 100 °C and saRNA-GFP + C protein showed round structures which are reported to exist only in the presence of binding nucleic acids. However, only in the saRNA-GFP + C protein sample were capsid-like structures with higher electron densities in the center of the particles identified (Figure 5).

To check if the association of saRNA-GFP and C protein could be delivered to cell lines, BHK-21 and J774A.1 cells were incubated with the complex (Figure 6).

The highest fluorescence intensity and best distribution was found in BHK-21 cells transfected with saRNA-GFP using the transfection reagent. However, when the saRNA-GFP alone or associated with C protein was delivered without the transfection reagent, no fluorescence was produced, showing that neither the association nor RNA only is able to enter BHK-21. When delivered to J774A.1, saRNA-GFP plus transfection reagent showed very limited fluorescence, indicating that macrophages are harder to transfect than BHK-21. However, a few fluorescence points could be distinguished in J774A.1 cells incubated with the saRNA-GFP + C protein, showing that the saRNA-GFP could be delivered to some cells.

## 3. Discussion

For many years, *Alphavirus* replicons have been considered good candidates for RNA-based vaccines. The development of new technologies aiming proper RNA assembly and delivery are critical steps for the proposition of vaccine candidates using self-amplifying RNA technologies. The objective of this work was to express, purify and use C protein from *Semliki Forest* virus for the formation of a saRNA/protein structure similar to *Alphavirus* capsid. Capsid structures have been engineered for improved saRNA stability [20] and besides that, they may present good properties on saRNA delivery, rapidly releasing saRNAs in the cytoplasm after cell uptake. Here, results showed that an SFV C protein was successfully expressed and purified from *E. coli*, preserving the saRNA binding property. The fast two step purification protocol established was able to separate the 33 kDa protein from a contaminant truncated form of 20 kDa. Further improvements on the C gene sequence are advisable to obtain only the expected protein, without generating the truncated form, probably a consequence of depletion in bacteria of charged lysine-transfer RNA (tRNA)Lys-UUU due to many repetitions of the same lysine codon. Other works with *Alphavirus* capsid proteins have shown that these proteins have preference for binding the viral RNA through s stem–loop structure, whose mechanism is conserved for other *Alphavirus* capsid proteins [21], but have no specificity for their parental RNA, binding to single-strand nucleic acids of various sizes [16,22] being mainly dependent on the positively charged residues at the N-terminus of the protein [23]. However, although in other works the C protein was also produced in bacteria and purified with similar strategies [16], no RNA contamination was observed in the purified protein. In this work, as RNA contamination was evidenced in the protein expressed at 28 °C and in fewer amounts in the protein expressed at 37 °C, and considering that basal RNA contamination could impair the association of C protein with saRNAs, the purified C protein was incubated at high temperatures as a method for breaking the association between C protein and any residual contaminant nucleic acid before the inclusion of the saRNA-GFP. Results did not confirm that heat treatment could break any pre-existing association, instead showing that when the C protein was heated and slowly cooled, the preparation presented capsid-like structures, which are reported as nucleic acid-dependent [24]. Where the capsid-like structures were present in the purified protein sample, disassembled during heat treatment and reassembled with remaining nucleic acids or just not disassembled during the treatment is unknow. However, when C protein was cooled and immediately added of saRNA-GFP, capsid-like structures with an electron-dense core were obtained, probably because of the strong biding property between C protein and the packaging signal present at the nsp2 [25]. These structures were exceptionally similar to the structures obtained in a study in which native C protein was associated with RNA [22]. The presence of many lysine residues, which are characteristic of nucleic acid binding proteins, preserved the capacity of the heat-treated C to bind the saRNA-GFP and to form capsid-like structures. This property can be valuable in further developments in a formulation of saRNA + C to be tested in vivo when the absence of bacterial nucleic acid will certainly be required. First assays on the delivery of saRNA-GFP + C protein to BHK-21 and to J774A.1 macrophages showed that only in the macrophage lineage did some cells present gene expression (GFP positive). As the BHK-21 cell line is commonly used for checking replicon activity through replication defective SFV infection or direct delivery [26,27], the lack of GFP expression in this case showed that the association saRNA-GFP + C protein is uncapable of entering in this cell line. On the other hand, macrophages can present in vitro endocytosis, which may be responsible for the saRNA-GFP deliver and GFP expression in some of the J774A.1 cells. This suggests that the complex can be part of a delivery system of a functional saRNA.

## 4. Materials and Methods

### 4.1. Cells, Culture Medium and Reagents

*Escherichia coli* BL21 (DE3) were cultured in LB medium (Thermo Fisher, Waltham, MA, USA) with 1.5% agar (Thermo Fisher) and 50 µg/mL kanamycin (Sigma, Kawasaki, Japan) for clone selection. For protein expression, *E. coli* were cultured in LB medium with 50 µg/mL kanamycin (LB/Kan). Samples were stored at −20 °C until analysis. A BL21 work bank was established with the selected clone in LB containing 20% glycerol and frozen immediately in liquid nitrogen before stocking at −80 °C. BHK-21 [C-13] from the laboratory work bank were cultured in alpha modified Minimum Essential Medium (αMEM) with 10% Fetal Bovine Serum (FBS) at 37 °C, J774A.1 (ATCC, Manassas, VA, USA) murine macrophages were kindly provided by the Laboratory of Bacteriology (Instituto Butantan) and were cultured in RPMI Medium (Gibco, Billings, MT, USA) with 10% FBS at 37 °C. Delivery assays were performed in 6-well plates containing 5 × 10^4^ cells/cm^2^.

### 4.2. Plasmid Construction

The cDNA corresponding to the *Semliki Forest* virus C protein (NC_003215.1) was synthesized as a double-strand fragment (gBlocks^®^ Gene Fragments, IDT Technologies, Coralville, IA, USA) after codon optimization for expression in prokaryotes, insertion of restriction enzyme and amplification sites. The cDNA was cloned on a pET28a(+) (Novagem, New York, NY, USA) after digestion with NcoI and EcoRI (Thermo Fisher), fragment purification from agarose gel (Wizard^®^ SV Gel and PCR Clean-Up System, Promega, Madison, WI, USA) and ligation with T4 DNA ligase (Thermo Fisher), following manufacturer instructions. *E. coli* DH5α were heat-shock transformed with the construction and selected for kanamycin resistance. Construction was first verified by restriction enzyme pattern and further by sequencing. The final construction encoded a N terminal six-histidine tag followed by C protein cDNA under the control of T7 promotor and was used to transform BL21 bacteria for protein expression.

### 4.3. SDS-PAGE and Western Blotting

Protein expression analysis was performed on 12% SDS-PAGE after diluting samples in Laemmli or lysis buffer. Samples with the same quantity of bacteria were centrifuged at 2600× *g* for 5 min, suspended in 50 µL lysis buffer (62.5 mM Tris, 2% SDS, 10% glycerol and 5% β mercaptoethanol) and then boiled at 100 °C for 10 min. Samples from purification steps were diluted (1:4) prior to loading buffer addition and heating. Electrophoresis was performed for three hours at 150 V. Gels were stained with PageBlue^TM^ Protein Staining Solution (Thermo) and protein sizes evaluated in relation to protein molecular weight marker (Page Ruler, Thermo Fisher). For Western blotting analysis, proteins were transferred to nitrocellulose membranes (Amersham Protran 0.45 NC, GE Healthcare, Chicago, IL, USA) at 12V for one hour. Membranes were then blocked with 3% low-fat milk in PBS, washed and further incubated with mouse anti-6x histidine monoclonal antibody (Cusabio, Houston, TX, USA), followed by incubation with goat anti-mouse HRP antibody (Sigma). Membranes were revealed with SuperSignal™ West Pico Chemiluminescent Substrate (Thermo Fisher) and documented in a chemiluminescence system (Alliance, Atlanta, GA, USA).

### 4.4. Protein Expression and Purification

A 100 µL work bank aliquot was inoculated in 15 mL LB/Kan and incubated at 37 °C at 240 rpm for 16 h. This culture was used to inoculate 250 mL LB/Kan and cultured under the same conditions until reaching an optical density of 0.6 at 550 nm. The temperature was maintained at 37 °C or lowered to 28 °C, and 1 mM IPTG (Sigma) was added. After 3 h or 6 h of induction (37 °C or 28 °C, respectively), the culture was centrifuged at 2600× *g* for 5 min and the pellet stored at −20 °C. For bacteria disruption, cell pellets were lysed in an ice bath by sonication with 60% amplitude, with 3 pulses of 4 s with 4 s intervals (Branson Sonifier 250A, Fort Lauderdale, FL, USA) in 30 mL lysis/equilibration buffer pH 7.4 (PBS, 500 mM NaCl, 10 mM imidazole, 1 mM PMSF) and further filtered at 0.45 µm. For C protein purification, samples of lysates were injected into previously equilibrated 1 mL HisTrap^TM^ FF (GE Healthcare) with a peristaltic pump at a flow rate of 0.5 mL/min. Subsequently, the column was washed with 10 CV of equilibration buffer and 2 CV of washing buffer (PBS, 500 mM NaCl, 70 mM imidazole) and eluted with 2 CV elution buffer pH 7.4 PBS-NaCl containing 800 mM or 1200 mM imidazole. Elution buffer was immediately exchanged for Hepes 25 mM, 133.6 mM NaCl, 1.7 mM MgCl, pH 7.4 with HiTrap^TM^ Desalting 5 mL (GE Healthcare). The protein was stored at 4 °C prior to further analysis and experiments. Protein concentration was measured using Pierce^TM^ BCA Protein Assay (Thermo) following manufacturer instructions.

### 4.5. In Vitro RNA Transcription

*Alphavirus* self-amplifying RNA was obtained from pSFVGFP as previously described [27]. Briefly, this plasmid contains all the non-structural proteins (nsp1-4) and the strong 26S subgenomic promoter from the *Semliki Forest* virus. Structural protein genes (C, E1, E2, E3) were removed, and in their place was inserted the gene of the green fluorescent protein (GFP) under the control of the 26S promoter (Figure 1). By in vitro transcription, this vector can generate a self-amplifying RNA (saRNA) containing all non-structural SFV coding genes besides the GFP gene. When delivered to a competent cell, the saRNA is translated into viral non-structural proteins which will produce more RNA copies as template for transcription of the gene of interest. Mayaro virus E2 coding RNA was obtained from pGEM^®^-T Easy (Promega) containing the cDNA of Mayaro Envelope 2 protein (NC_003417.1), obtained from the cloning of a custom-synthesized gBlock (IDT Technologies). Briefly, 8 µg plasmids were linearized with 10 U *Nru* I or *Sca* I, respectively, and submitted to in vitro transcription at the concentration of 1 µg/µL, using MegaScript SP6 (Thermo Fisher) at 40 °C in the presence of RNase inhibitor (Ribolock, Thermo Fisher) for two hours. After the reaction 4 U of Turbo DNase (Thermo Fisher) were added for DNA removal and RNA samples were incubated at 37 °C for 15 min. After quantification by fluorometry (Qubit, Thermo Fisher) RNA were immediately used in association experiments.

### 4.6. Association between C Protein and RNA

For the association of C protein and RNA, the molecular proportion was established at 400 C protein molecules to one RNA molecule, that is, 4.13 µg C protein to 1 µg saRNA-GFP or 12.8 µg C protein to 1 µg MAYV_E2 RNA. Then the protein and RNA were diluted in 25 mM Hepes buffer pH 7.4 and incubated at 30 °C for 30 min. Alternatively, the C-purified protein was first heated at 70 °C or 100 °C for 10 min and cooled slowly before being associated with RNA to break the binding of contaminant nucleic acids to the C protein. The complexes resulting from protein C/RNA association were analyzed by mobility shift assay in 0.8% agarose gel. Samples and controls were first mixed with GelRed^TM^ nucleic acid gel stain (Biotium, Fremont, CA, USA) and then electrophoresis was applied at 70 V for 90 min. Results were documented using the UV fotodocumentation system (Alliance). Complexes were also analyzed by transmission electron microscopy (TEM) after negative staining of 5 µL samples in carbon coated grid with 2% uranyl acetate. Samples were visualized in a LEO 906E microscope (Zeiss, Jena, Germany).

### 4.7. Delivery of Protein C/RNA to Mammalian Cells

Protein C/RNA complexes were prepared for delivery to BHK-21 or J774A.1 cells and diluted in serum-free αMEM or RPMI, respectively. Samples containing complexes, previously checked for cytotoxicity absence, were placed on 6-well plates containing BHK-21 or J774A.1 at 37 °C for 3 h and 5% CO_2_. Positive control was prepared with in vitro transcribed saRNA-GFP mixed with Transmessenger (Qiagen, Venlo, The Netherlands) transfection reagent following manufacturer instructions. Cells were monitored for GFP expression by fluorescence microscopy in BX51 fluorescence microscope (Olympus, Hamburg, Germany), with assistance from DP73 image capture system (Olympus) and CellSens Standard V 1.7 software.

## Data Availability

Original data is available upon a reasonable request.

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
