# Peer review of "Semliki Forest Virus (SFV) Self-Amplifying RNA Delivered to J774A.1 Macrophage Lineage by Its Association with a Purified Recombinant SFV Capsid Protein"

_ijms, 2024, doi:10.3390/ijms25147859_

Round 1

Reviewer 1 Report

Comments and Suggestions for Authors

authors describe the bacterial recombinant production and His-tag purification of SFV capsid and its complexation with reporter RNA. They then assess the potential of the resulting complexes to allow reporter gene expression in cell culture. 

authors state that the reason for testing a couple of methods to remove residual RNA from the purified recombinant capsid protein was the detection of this nucleic acid in the staining of the retardation assay. However there is no such discernible signal (Fig3 lane2). 

Nevertheless they continued with treating the capsid protein with RNase, reducing agents, detergents and heat. This was described in the discussion section but would fit better in the results section. 

Without any elaboration on the methods/materials they used (which RNases, detergents,..?) they state that heating to temperatures of 70-100°C was most effective in removing the contaminating nucleic acids. Clearly one has to wonder how they measured this as no contaminating nucleic acid is detectable in the retardation gel and no other means of analysis is mentioned. 

Authors should show these data. Namely, the capsid treated with the different methods (RNase, detergents, reducing agents) and their 1) binding of residual contaminant nucleic acids and their 2) binding to target RNA, in the retardation assay. 

authors do not include a control condition where purified capsid without heat-treatment is allowed to bind to the target RNA in the retardation assay. As such it seems that the target RNA only binds to the heat treated capsid. 

It is very hard to believe that heating the purified protein to 70-100°C allows it to maintain its soluble native folding. While a EM snapshot is provided of two particles showing a capsid-like shape this is hardly convincing as no further sampling of EM structures seemed to have been performed and medium size distributions calculated as is common. This should be included. 

Comments on the Quality of English Language

some minor grammatical and punctuation corrections

Author Response

authors describe the bacterial recombinant production and His-tag purification of SFV capsid and its complexation with reporter RNA. They then assess the potential of the resulting complexes to allow reporter gene expression in cell culture. 

 The authors are very grateful for the observations made in the manuscript. We revised it and the main modifications are colored in red. We also improved the terminology as from the reviews we could notice that the text was not clear as it should be. All “replicons” and “SFV RNA” are now referred as self-amplifying RNA or “saRNA”. When this is the RNA used in the work, coding for the GFP, it was termed as “saRNA-GFP” and finally, its association with the C protein ir referred now as “saRNA-GFP + C protein”.

authors state that the reason for testing a couple of methods to remove residual RNA from the purified recombinant capsid protein was the detection of this nucleic acid in the staining of the retardation assay. However there is no such discernible signal (Fig3 lane2). 

Thank you for your observation. In fact we have seen more or less contamination depending on the process. As it is still not a very standardized method of protein production this can be solved in future work. We added an observation about the sensitivity of the methodology used for assessing this contamination. Although it is not seen in this gel as pointed out, the contamination can still be present as it is the nature of the C protein to bind to nucleic acids. We also introduced one more observation about this and the consequence that can be verified by the formation of nucleocapsid like structures in the absence of added RNA, a phenomenon that is considered not possible by distinguished authors:

“When the C protein was heated and slowly cooled it demonstrated the property of association with the remaining nucleic acids, as the formation of capsid-like structures are assumed as nucleic acid dependent (Tellinguisen et al., 2000). When it was cooled and immediately added of saRNA-GFP, capsid-like structures with an electron dense core were obtained, probably because of the strong biding property between C protein and the packaging signal present at the nsp2 (Mendes and Kuhn, 2018)”.

Nevertheless they continued with treating the capsid protein with RNase, reducing agents, detergents and heat. This was described in the discussion section but would fit better in the results section. 

The mention was removed from the discussion. The RNase treatment was done as a strategy to remove de RNA, but at it cannot be removed itself after the reaction, we would not be able to associate a saRNA to the C protein, so it was just tested in one occasion and this is not relevant to the article.

Without any elaboration on the methods/materials they used (which RNases, detergents,..?) they state that heating to temperatures of 70-100°C was most effective in removing the contaminating nucleic acids. Clearly one has to wonder how they measured this as no contaminating nucleic acid is detectable in the retardation gel and no other means of analysis is mentioned. 

Comments about the reason why we tried the incubation as a method to remove the nucleic acid contamination were included in the results section.

Authors should show these data. Namely, the capsid treated with the different methods (RNase, detergents, reducing agents) and their 1) binding of residual contaminant nucleic acids and their 2) binding to target RNA, in the retardation assay. 

authors do not include a control condition where purified capsid without heat-treatment is allowed to bind to the target RNA in the retardation assay. As such it seems that the target RNA only binds to the heat treated capsid. 

A new gel image (new figure 3) was introduced to show this important aspect brought by the reviewer. In this gel it is shown the association of saRNa with C protein not heat treated. Also it is shown the association of other RNA type with C protein showing that this association is not specific and although there is a nucleic acid contamination, there is still biding capacity available in the purified C protein material.

It is very hard to believe that heating the purified protein to 70-100°C allows it to maintain its soluble native folding. While a EM snapshot is provided of two particles showing a capsid-like shape this is hardly convincing as no further sampling of EM structures seemed to have been performed and medium size distributions calculated as is common. This should be included. 

A new photomicrography was included showing the capsid-like structures in the absence of the saRNA. These capsid structures were obtained after incubating the protein at 100 °C for just 10 minutes. 

Reviewer 2 Report

Comments and Suggestions for Authors

In the manuscript by Gomes et al., the authors purify SFV C protein and show that it associates with SFV GFP-tagged RNA and can be delivered to J7744A.1 murine macrophages.  In general, the manuscript is briefly written and more details are necessary for the reader to fully understand the results.  Below are my suggestions for improvement:

Lines 104-106 and Fig 4 - an experiment must be performed here with the absence of SFV RNA.  Will the C protein still self-assemble into capsids?  More explanation of the experiment is also needed in the text.

Lines 107-109 and Fig 5 - this experiment does not check if the capsid-RNA complex "could be undone".  The experiment determines if the C protein is delivered to the cells.

More explanation of why the experiment fails in BHK21 cells is needed.  Can SFV infect BHK21 cells?  If so, the results shown here do not make sense.  

What exactly is the SFV RNA that is being used?  A schematic should be shown.  Which SFV gene is it?  If it is full-length SFV RNA, which also encodes for the capsid gene, then the experimental design is not correct since the RNA can also make capsid protein itself.

Line 144 - what is "GPV?"

What are J7744A.1 cells?  Is this a typo?

Author Response

In the manuscript by Gomes et al., the authors purify SFV C protein and show that it associates with SFV GFP-tagged RNA and can be delivered to J7744A.1 murine macrophages.  In general, the manuscript is briefly written and more details are necessary for the reader to fully understand the results.  Below are my suggestions for improvement:

The authors are very grateful for the observations made in the manuscript. We revised it and the main modifications are colored in red. We also improved the terminology as from the reviews we could notice that the text was not clear as it should be. All “replicons” and “SFV RNA” are now referred as self-amplifying RNA or “saRNA”. When this is the RNA used in the work, coding for the GFP, it was termed as “saRNA-GFP” and finally, its association with the C protein ir referred now as “saRNA-GFP + C protein”.

Lines 104-106 and Fig 4 - an experiment must be performed here with the absence of SFV RNA.  Will the C protein still self-assemble into capsids?  More explanation of the experiment is also needed in the text.

A new result was included and more information provided. It was documented in different studies that the C protein from Alphavirus cannot form capsid-like structures without the association with a nucleic acid. In our case we found this structures before the introduction of the self-amplifying RNA, suggesting that the purified protein was still contaminated with some RNA from bacteria.

Lines 107-109 and Fig 5 - this experiment does not check if the capsid-RNA complex "could be undone".  The experiment determines if the C protein is delivered to the cells.

This is correct, thank you for the observation. Changes were made in the text.

More explanation of why the experiment fails in BHK21 cells is needed.  Can SFV infect BHK21 cells?  If so, the results shown here do not make sense.  

More information was added in this aspect. The C protein associated with RNA is not capable itself of entering the cells. Normally this could be explained by the surface charges of the particle and the cell. In this case the other structural proteins of SFV are absent, and this is critical for the RNA delivery because they are the responsible for receptor binding and fusion of the virus particle. This is the reason why BHK-21 cells which are normally and easily infected by SFV and other alphaviruses cannot be “infected” by the non-enveloped capsid.

What exactly is the SFV RNA that is being used?  A schematic should be shown.  Which SFV gene is it?  If it is full-length SFV RNA, which also encodes for the capsid gene, then the experimental design is not correct since the RNA can also make capsid protein itself.

Scheme 1 was introduced. Authors consider that this made good improvement to the understanding of the replicon and its mode of action.

Line 144 - what is "GPV?"

This was a typo.

What are J7744A.1 cells?  Is this a typo?

This was a typo.

Round 2

Reviewer 1 Report

Comments and Suggestions for Authors

authors clarified some statements and added some data. 

They do however again fail to provide evidence that their heat treatment removed the residual RNA, if there was any in the first place, or had any impact on downstream application. 

They reference reports from the Kuhn group but do not mention that these researchers, despite using similar protein purification strategies, do not witness residual RNA binding or any issue with binding to subsequently added RNA without heat- or other treatment (PMID: 10364277). The authors should discuss the discrepancies between the studies. 

Author Response

Dear reviewer, thanks again for the observations. We hope to have clarified them with the following explanations and alterations in the paper.

They do however again fail to provide evidence that their heat treatment removed the residual RNA, if there was any in the first place, or had any impact on downstream application. 

The heat treatment did not removed any residual RNA as correcly stated by the reviewer. We changed this terminology. The heat treatment instead could have broken the association between any residual RNA and the purified C protein. But this is not demonstrated by the results as observed. The results show that the C protein treated at high temperatures is still capable to associate with nucleic acids but does not show the benefits of this approach. These considerations were included in the paper.

They reference reports from the Kuhn group but do not mention that these researchers, despite using similar protein purification strategies, do not witness residual RNA binding or any issue with binding to subsequently added RNA without heat- or other treatment (PMID: 10364277). The authors should discuss the discrepancies between the studies.

The mention was included. We revised this new version to be clear that the heat treatment didn't show benefits in terms of purity or better association property. A consideration was made in the discussion section but no extensive hypothesis were raised to explain the discrepancies between the studies because this could bring another questions which would necessarily need more experimentation to be solved, as they can be related to buffer differences, sequences, bacteria strains, culture or lysis conditions. The reviewer adverted in the first report that the amount of RNA which could be contaminating the protein was undetectable or not proved in the first version of the paper. This new version states clearly that there is no sign of RNA contamination by the analysis of the gel, but the RNA contaminantion cannot be excluded having in mind the limit of detection of the method used for analysis and the formation of capsid-like structures identified in the transmission eletronic microscopy. This consideration was also included in the paper.

Reviewer 2 Report

Comments and Suggestions for Authors

The authors have adequately addressed my previous concerns.

Author Response

Thank you for your revision. New modifications were added in response to the other reviewer concerns.